# Effect of Adding Different Commercial Propylene Glycol Alginates on the Properties of Mealworm-Flour-Formulated Bread and Steamed Bread

**DOI:** 10.3390/foods12193641

**Published:** 2023-10-01

**Authors:** Xinyuan Xie, Xiaolong Zhao, Fanjian Meng, Yupeng Ren, Jianhui An, Lingli Deng

**Affiliations:** College of Biological and Food Engineering, Hubei Minzu University, Enshi 445000, China; 202130346@hbmzu.edu.cn (X.X.); 202240519@hbmzu.edu.cn (X.Z.); 202240547@hbmzu.edu.cn (F.M.); 202230344@hbmzu.edu.cn (Y.R.); 2020049@hbmzu.edu.cn (J.A.)

**Keywords:** propylene glycol alginate, mealworm, bread, steamed bread, texture

## Abstract

Mealworm-flour-formulated flour-based products have gained increasing attention; however, their textural properties need to be improved. Propylene glycol alginate (PGA) is a commercial food additive with excellent emulsifying and stabilizing capabilities. We evaluated the effects of adding three commercially available PGAs (0.3% *w/w*, as food additive) on the properties of 10% concentration of mealworm-flour-formulated bread and steamed bread. The results showed that, compared with the control (2.17 mL/g), three PGA brands (Q, M, and Y) significantly increased the specific volume of the bread to 3.34, 3.40, and 3.36 mL/g, respectively. Only PGA from brand Q significantly improved the specific volumes of bread and steamed bread. The color of the bread was affected by the Maillard reaction. The addition of PGAs also augmented the moisture content of the fresh bread crumbs and steamed bread crumbs. All three PGAs improved the textural properties of bread and steamed bread. During storage, PGA addition delayed the staling of bread and steamed bread. In summary, our study showed that the addition of 0.3% PGA from three different producers improved bread properties, with PGA from brand Q having the most substantial effect. PGA had a more substantial effect on bread than steamed bread. Our results provide a theoretical basis to guide the development of insect-formulated flour-based products.

## 1. Introduction

Insect food has emerged as a new trend in protein alternatives and supplements [1]. Insects have been part of the Chinese diet for over 2000 years, with a rich history of consumption. Because they are rich in protein, edible insects are considered healthier and a more sustainable alternative to traditional meat sources, and therefore, they have become increasingly available in the Chinese marketplace [2]. The yellow mealworm (*Tenebrio molitor*) has gained attention in recent years owing to its nutritional composition, accessibility, and legal approval, according to Regulation (EU) 2015/2283 [3]. Numerous studies have investigated the use of mealworm flour in foods such as biscuits [4], breads [5,6], and meat [7,8]. Mealworm flour supplementation can enrich the nutritional properties of flour-based products owing to its protein and lipid content, which can be up to 50% and 28%, respectively [9,10]. However, mealworm flour can deteriorate the textural properties of flour-based products. Our previous study found that low mealworm substitution levels (5% and 10%) led to higher specific volume and reduced bread hardness, whereas higher levels of substitution (15% and 20%) resulted in a marked reduction in bread quality [5]. A similar dose-dependent effect of substitution was observed by Roncolini et al. [6], who found the highest specific volume at a substitution level of 5% and a decrease at 10% substitution. According to Khuenpet et al. [11], the specific volume of bread gradually decreased with increasing levels of mealworm substitution (by up to 15%), and bread hardness increased by approximately four times compared with the control bread. In contrast, González et al. [12] observed a significant decrease in specific volume when 5% wheat flour was replaced with mealworm flour. 

Hydrocolloids are often used to enhance dough consistency and air bubble entrapment capacity in flour-based products [13,14]. Among the hydrocolloids used in bakery products, propylene glycol alginate (PGA) is commonly used because of its lipophilic and hydrophilic properties, which provide excellent emulsifying and stabilizing effects. Propylene glycol alginate (PGA) is an alginate derivative formed via esterification reaction between alginic acid and propylene oxide [15]. According to the GB 2760 China Food Additive Use Standard, PGA can be used as a thickener, emulsifier, and stabilizer. Moreover, expansion of its application to bakery food was approved in 2017. Tabara et al. [16] found that the addition of PGA improved bread height and specific volume. PGA has also been used to develop gluten-free bread [13]. Zhao et al. [17] reported that the addition of PGA and hydroxypropyl methylcellulose increased the density of gluten-free bread. Alamri et al. [14] developed potato gluten-free doughs and found that potato gluten-free doughs with binary blends of flaxseed gum and PGA exhibited higher viscoelasticity than that of other doughs. However, few studies have investigated the effects of different commercial PGAs on the textural properties and staling of bread and steamed bread.

This study hypothesized that different commercial PGAs could improve the properties of mealworm-flour-formulated bread and steamed bread, providing evidence supporting the use of PGA in bread production. Specific volume, porosity, colorimetric, and textural analyses of bread and steamed bread were conducted. To further investigate the influence of PGA on the staling of bread and steamed bread, texture measurements were performed to analyze starch retrogradation during storage. This study aimed to provide a theoretical basis for selecting appropriate PGA raw materials for the development of insect-formulated flour-based products.

## 2. Materials and Methods

### 2.1. Materials

High-gluten wheat flour (protein content: 12.8%) and whole wheat flour (protein content: 12.5%) were supplied by Xinxiang Xinliang Cereals Processing Co., Ltd., Xinliang, China. The mealworm flour (protein content: 43.5%, fat: 25.3%, carbohydrate: 1.17%, dietary fiber: 22.1%, energy: 406.38 kcal/100 g, Na: 1.3 × 10^3^ mg/kg) was supplied by Qingdao Sino-Crown Biological Engineering Co., Ltd. (Qingdao, China) [4]. Butter and yeast were supplied by Angel Yeast Co., Ltd. (Wuhan, China). Milk powder, salt, and sugar were purchased from a local grocery store. PGA (Q, degree of esterification ≥ 80%) was purchased from the Qingdao Bright Moon Seaweed Group Co., Ltd. (Qingdao, China) [17]. PGA (M, purity of 98%) was purchased from Shanghai Macklin Biochemical Co. Ltd. (Shanghai, China). PGA (Y, degree of esterification 75%, purity of 98%) was purchased from Yuanye Biotechnology Co., Ltd. (Shanghai, China) [18,19].

### 2.2. Preparation of Wheat Flour Formulated with Mealworm Flour

To create composites of wheat/whole wheat flour and mealworm flour, mealworm flour was used to replace wheat/whole wheat flour at a weight ratio of 10%.

### 2.3. Bread Preparation 

Bread production was performed as previously described [5]. The bread was prepared following the GB/T 14611–2008 Straight dough method with modifications [5]. Figure 1 depicts the method used to produce mealworm-flour-formulated bread. The formulations are listed in detail in Table 1. PGA was added to the mixed flour at a concentration of 0.3% (*w/w*). Bread without PGA was used as a control (CB). The raw materials were mixed using a dough mixer (AM-CG108-1: ACA, North America Electric Appliances (Zhuhai) Co., Ltd., Zhuhai, China) for 20 min. The resulting dough was then leavened at 30 °C in a fermentation cabinet (DHTHM-16-0-P-SD, Doaho Test Co., Ltd., Shanghai, China) for 70 min. After fermentation, the dough was divided into approximately 75 g pieces, shaped into round shapes, and proofed at 30 °C for 45 min. Finally, the bread was baked at 175 °C using top and bottom heat in a steam oven (K6: Daewoo, Incheon, Republic of Korea) for 20 min. After baking, the breads were cooled in air for 1 h. Then, the cooled samples were stored in hermetically sealed bags for evaluation or at room temperature for 0–3 days.

The bread was configured with PGA from different manufacturers and named QB, MB, and YB.

### 2.4. Steamed Bread Preparation

The method used to prepare (SB) is shown in Figure 1. These formulations are listed in Table 2. PGA was added to the mixed flour at a concentration of 0.3% (*w/w*). Steamed bread without PGA was used as a control.

A yeast mixture was prepared by mixing yeast (3 g) in water (50 mL). All raw materials, yeast liquid, and additional water (160 mL) were mixed in a dough mixer (AM-CG108-1: ACA, North America Electric Appliances (Zhuhai) Co., Ltd., Zhuhai, China) until the gluten was expanded. Subsequently, the dough was removed and fermented in a fermentation cabinet (DHTHM-16-0-P-SD; Doaho Test Co., Ltd., Shanghai, China) for 60 min at 30 °C and at 85% relative humidity. The fermented dough was kneaded manually into the exhaust gas and sheeted with a rolling pin. The dough sheet was rolled into a cylinder and cut into a suitably sized steamed bread dough before further fermentation for 20 min. The resulting steamed bread dough was steamed in a steam oven (K6: Daewoo, Incheon, Republic of Korea) for 30 min at 105 °C. Finally, the steamed breads were cooled at room temperature for 1 h for evaluation or packed in hermetically sealed bags and frozen at −18 °C for 0–3 days. 

### 2.5. Colorimetric Analysis of Bread and Steamed Bread

We referred to our previous experimental methods for color measurements [5]. The color of the bread crumbs (each 2.0 cm thick) and steamed bread (each 1.5 cm thick) was measured using a colorimeter (CS-820N, Hangzhou CHNSpec Technology Co., Ltd., Hangzhou, China). This determination was based on three analyses of the *L**, *a**, and *b** color system.

### 2.6. Specific Volume Analysis of Bread and Steamed Bread

Volumes of bread and steamed bread were measured using the millet displacement method described by Xie et al. [5]. The formula for the calculation is as follows:Specific volume (mL/g)=volume (mL)/weight (g)

### 2.7. Porosity Analysis of Bread and Steamed Bread

Crumb porosity was estimated using ImageJ software (US National Institutes of Health, Bethesda, MD, USA). According to Kowalski et al. [9], image analysis was employed to assess cell density (crumb porosity). ImageJ software was used to analyze the internal texture structure of the bread after obtaining a color image of the cross-section. To acquire the cross-sectional porosity, the image was transformed into an 8-bit grayscale image and the corresponding region was selected for investigation.

### 2.8. Texture

#### 2.8.1. Texture Analysis of Bread

To perform texture profile analysis (TPA) of the crumb, a TA-XT Plus texture analyzer (Shanghai Baosheng Industrial Development Co., Ltd., Shanghai, China) equipped with a P/36R cylinder probe was used, following a previously described method. The bread was stored at room temperature for 0–3 days. Bread slices (each 2.0 cm thick) were cut from the center of each bread.

The test parameters were as follows: deformation of 40% in two subsequent cycles, pre-test speed of 3 mm/s, test and post-test speeds of 1 mm/s, and a trigger force of 5 g. The crumbs were evaluated in terms of several parameters, including hardness, springiness, chewiness, gumminess, cohesiveness, and resilience.

#### 2.8.2. Texture Analysis of Steamed Bread

The steamed breads stored at −18 °C for 0–3 days were re-steamed at 105 °C for 30 min and kept an air environment for 1 h.

The textural properties of the steamed bread crumbs were studied using the same TA-XT plus texture analyzer with a cylinder probe (P/36). Steamed bread (1.5 cm) was cut for testing. 

In the texture profile analysis mode, the pre-test speed was 3 mm/s, and the test and post-test speeds were both 1 mm/s, with a delay of 2 s between the first and second compressions. A trigger force of 10 g was used to compress steamed bread slices to 50% of their original height. Hardness, springiness, chewiness, gumminess, cohesiveness, and resilience were recorded. 

### 2.9. Moisture Content of Fresh Bread Crumb and Steamed Bread Crumb

According to American Association of Cereal Chemists (AACC) Method 44-15A [20], the moisture contents of fresh bread and steamed bread were determined using the air-oven method. The moisture content of the bread crumb and steamed bread crumb were determined by oven drying for 24 h at 105 °C. The bread slices (2.0 cm thickness) were cut from the central portion of the bread and steamed bread, and a crumb square (1.0 cm length) was taken from the center of each slice and used for moisture determination.

### 2.10. Statistical Analysis

All tests were performed at least in triplicate. Values are presented as arithmetic means ± standard deviation (SD), and Origin 8.0 software (OriginLab, Northampton, MA, USA) was used to analyze the results. The data were analyzed using an analysis of variance (ANOVA), and Tukey’s test was performed for comparative analysis at a significance level of 95% (*p* < 0.05). 

## 3. Results 

### 3.1. Bread Characteristics

#### 3.1.1. Specific Volume, Color, Porosity, and Moisture Content

One of the key factors determining bread quality and acceptability is color, which is determined by moisture content, the Maillard process, and caramelization during baking [14]. The bread, crumb, and binarized images of the cross-section are presented in Figure 2, and the *L**, *a**, and *b** color parameters of the bread crumbs are presented in Table 3. Color analysis showed that the color of the bread crumbs was not significantly affected by the addition of different PGAs.

Specific volume, which refers to the dough’s ability to inflate during baking, is a crucial characteristic of bread quality [21]. The use of PGA had a positive effect on the bread specific volume. Table 3 demonstrates that PGA significantly increased the specific volume of bread compared to formulations without PGA. The specific volume of bread significantly increased from 2.17 mL/g (CB) to 3.40 mL/g (MB).

Breads with high porosity and specific volume are generally thought to have better baking quality. The porosity results of the bread crumbs are presented in Table 3 and Figure 2. Compared to the control bread, the porosity of MB decreased, although there was no discernible difference. QB and YB exhibited the highest porosities of 31.91% and 34.15%, respectively. These results indicated that most PGA optimized bread porosity. 

The moisture contents of fresh mealworm-flour-formulated bread with different commercial PGAs are presented in Table 3. The experimental results showed that the moisture content of the fresh bread crumbs with PGAs added was consistently higher than that of the control bread, especially QB, which had a moisture content of 37.64%. 

#### 3.1.2. Texture Analysis

Bread texture is the most important quality indicator influencing consumer acceptance. Generally, there is a negative correlation between the hardness and bread quality [5]. Table 4 shows the textural characteristics of bread crumbs prepared using different commercial PGAs. Texture analysis indicated that the hardness of the bread sample containing PGA was significantly lower than that of the control sample. The hardness was affected by the addition of PGA, and the range varied between 4.57 N and 9.77 N for different samples. Sample YB had a minimum hardness of 4.57 N. Springiness is another important quality indicator. However, the samples did not exhibit significant differences in springiness. With the addition of PGA, the chewiness values were 5.81, 4.10, 3.53, and 2.90 N. Gumminess was significantly decreased (3.28 N) in the YB sample than in the control sample (6.64 N). PGA had a positive impact on cohesiveness and resilience, with the highest scores being 0.80 (QB) and 0.48 (MB), respectively. The effect of Brand Q PGA was statistically significant.

Measuring bread hardness on different storage days is important because increased hardness is often an indicator of staling [22]. Figure 3 and Table 4 show the hardness of bread crumbs during storage. Throughout the storage period, the hardness of the control bread was higher than that of the other bread samples. The hardness of bread with added PGA stored at room temperature for 3 d was lower than that of bread without PGA stored for 1 d. Therefore, we believe that the addition of PGA has a beneficial effect on the process of bread hardening. There was no significant change in the springiness of bread during storage. The chewiness and gumminess of bread showed a trend similar to that of hardness. In general, breads made with PGA showed similar chewiness and gumminess values after 3 d of storage, in contrast with breads made without PGA after 1 d of storage. The cohesiveness and resilience of bread slowly decreased during storage; however, the cohesiveness and resilience of QB, MB, and YB were higher than that of the control bread. It is possible that the addition of PGA slows down the bread staling process.

### 3.2. Steamed Bread Characteristics

#### 3.2.1. Specific Volume, Color, Porosity, and Moisture Content

Color is a crucial quality indicator for foods, including steamed bread, which is a staple food in the daily diet of Chinese residents. Color is a key factor in both consumption and popularity because it serves as an intuitive indicator of steamed bread [23]. Table 5 shows the color parameters (*L**, *a**, and *b**) of the crumbs in whole wheat flour steamed bread samples with different commercial PGAs. The appearance of steamed bread and crumbs is shown in Figure 4. The *L**, *a**, and *b** parameters showed no significant changes. Thus, the results indicated that PGA addition had no significant influence on the color of the steamed bread crumbs.

The desirable quality characteristics of steamed bread include a high specific volume, soft texture, and homogenous crumb grain structure [24]. Table 5 shows the specific volumes of the steamed bread, which ranged from 1.22 (MSB) to 2.23 mL/g (QSB). Notably, only Brand Q PGA significantly increased the specific volume of steamed bread. There was no significant difference between the CSB and YSB samples, whereas MSB showed the lowest specific volume (1.22 mL/g). No clear effect of PGA on the specific volume of steamed bread was observed.

Incorporating PGA into steamed bread samples resulted in a less homogenous crumb grain structure, which was reflected by lower porosity. The data in Table 5 show that the porosity of steamed bread was negatively affected by the addition of PGA. The porosity of the steamed bread samples decreased significantly, from 42.16% (CSB) to 29.63% (MSB).

The addition of PGAs also augmented the moisture content of the fresh steamed bread crumbs. The moisture contents of QSB (46.90%) and MSB (46.42%) were significantly higher than that of the control sample (45.47%). The moisture content of YSB was similar to that of the control group, but still higher than that of the control group.

#### 3.2.2. Texture Analysis

The texture of food is related to its sensory properties and is primarily defined by its tissue characteristics. The textural characteristics of steamed bread crumbs were affected by the addition of PGA (Table 6). These findings indicate that the addition of PGA gradually reduced the hardness of steamed bread, while marginally increasing its springiness, cohesiveness, and resilience. The hardness varied significantly (43.97–72.17 N) among samples and YSB had the lowest hardness (43.97 N), followed by samples QSB (65.33 N) and MSB (68.35 N). Springiness ranged from 0.70 to 0.91, with MSB exhibiting the maximum springiness and the control steamed bread exhibiting the minimum springiness, indicating that the increase in PGA in the formulation positively influenced the springiness of the steamed bread. The chewiness of the samples ranged from 21.91 to 45.27 N and was directly proportional to gumminess. QSB (0.76) and MSB (0.73) exhibited high cohesiveness, indicating an increase in the strength of the internal bonds between the steamed bread ingredients. The cohesiveness and resilience of steamed bread samples showed similar trends. PGA significantly enhanced the cohesiveness and resilience of steamed bread. Hence, we confirmed the beneficial effects of the addition of PGA on steamed bread properties.

The results in Table 6 show that the texture of the steamed bread changed during storage. Figure 5 and Table 6 show the changes in hardness of steamed bread during storage. The control steamed bread sample reached a hardness of 70.53 N after 3 d of storage. Meanwhile, the hardness of steamed bread with the addition of PGA was only 62.31 N (QSB), 45.85 N (MSB), and 49.43 N (YSB). These data indicate that PGA can effectively decrease the hardness of steamed bread and slow down the staling process. There were no significant differences in the cohesiveness between the samples during storage. Overall, the texture of steamed bread stored for 3 d showed little change, which may be related to the storage mode. The results showed that PGA greatly decreased staling by stabilizing the texture of steamed bread after storage [25].

## 4. Discussion

Assessing the color parameters (*L**, *a**, and *b**) is crucial, as bread color is a significant quality indicator. The color of bread samples can be affected by various factors, such as the ingredients used, their ratio in the recipe, and baking conditions. Additionally, the color value can impact the acceptability of bread to consumers [26]. Based on the experimental data, we found that PGA had no significant effect on bread color as a thickener, emulsifier, or stabilizer. Unlike bread, the Maillard reaction in steamed bread was not obvious during steaming. The color of steamed bread is influenced by the flour color. As with bread, PGA had no significant effect on the color of steamed bread. The addition of PGA significantly increased the moisture content of bread and steamed bread. This indicates that PGA can enhance the water retention capacity of bread and steamed bread, which was consistence with the decreased hardness.

PGA is the most frequently employed hydrocolloid for bread-making. Hydrocolloids are used to form a viscoelastic structure that compensates for gluten deficiency [13]. The surface activity and emulsifying properties of PGA are attributable to its unique combination of lipophilic and hydrophilic characteristics, making it a potential ingredient for enhancing bread texture and stabilizing air bubbles [17,27,28]. Therefore, PGA is an emulsifier that can increase the dough volume during fermentation [29]. This is because they can bind to water. PGA may delay the pasting and gelatinization of starch granules, resulting in an increase in starch gelatinization temperature. As a result, the viscosity is lowered and the bread volume increases owing to greater gas cell expansion [22,30]. Therefore, PGA improved bread quality with respect to specific volume and porosity (Table 3). Another possible reason is that some researchers found that the expansion of bread dough increased as a result of the addition of PGA to wheat flour and that the CO_2_ gas produced by the yeast could be efficiently trapped by the PGA wheat flour bread dough [16].

Our experimental results were similar to those reported by Zhao et al. [17]. They found that PGA increased loaf specific volume, with concentrations of 3.0% PGA and 2.0% PGA having the largest effect as well as a significant impact on the springiness of bread. Tabara et al. [16] investigated the impact of six different alginic acid-related materials on the quality of bread, specifically height (mm) and specific volume (cm^3^/g). They found that compared to the control sample, 2.0% (*w/w*) PGA significantly increased the height and specific volume of bread, indicating a positive impact of PGA addition on the quality of bread. 

The texture of bread crumbs is affected by the ingredients and recipes used. Peressini et al. [31] found that breads with PGA exhibited greater improvements in terms of increased specific volume, decreased bread crumb hardness, and improved bread crumb structure than breads made with xanthan gum. Xie et al. [32] discovered that the distribution of the wheat gluten network structure became denser, and the pore diameter decreased as the amount of γ-PGA increased, indicating that the addition of γ-PGA significantly enhanced the wheat gluten network.

Generally, bread crumb hardness is often used as an index of bread staling during storage [13,22,33]. The effect of PGA on the hardness of bread and steamed bread during storage should be investigated, as it is usually associated with changes in the starch fraction [34]. Based on the experimental results (Table 4), we inferred that PGA could decrease bread hardness and delay the staling of mealworm-formulated bread during storage. The most important causes of bread staling are often considered to be starch transformation, starch–gluten interactions, and moisture redistribution [25]. The hydrogen bonds formed between PGA and starch can also slow down starch retrogradation [13,35]. Shyu et al. [22] studied the effects of adding PGA at different concentrations (0.5, 1.0, and 5.0 g/kg, *w/w*) on the staling of bread. They revealed that bread containing 1.0 g/kg PGA had a lower hardness value after 5 d of storage than the control bread sample after only 1 d of storage. 

On the other hand, the effect of PGA addition on the specific volume, porosity, and texture of steamed bread differed from that of baked bread. This may be due to differences in the production of bread (baking) and steamed bread (steaming). The impact of incorporating PGA into steamed bread may differ from its effects in baked bread, owing to differences in the redistribution of water from hydrated proteins to native starch during the heating process. Steaming bread is characterized by simultaneous heat and water transport during the steaming process [36]. Gao et al. [37] observed the microstructures of baked and steamed bread using scanning electron microscopy and found that the internal structure of baked bread was more compact than that of steamed bread. The porosity of steamed bread was higher because steam heating allows starch granules to thoroughly absorb water and gelatinize properly, resulting in a more uniform gluten network [27,37]. The addition of a higher amount of steam during the steaming process can reduce the rate of water vapor transfer. As a result, starch gelatinization is promoted, leading to the formation of a less porous protein network [20]. PGA enhances the specific volume of bread by delaying starch gelatinization and increasing the gelatinization temperature. However, in the case of steamed bread, PGA addition did not significantly affect the specific volume. This may be because during the steaming process, steam promotes starch gelatinization, thereby counteracting the effects of PGA. In contrast, when using the baking method, the high temperature causes rapid evaporation of water; starch granules are unable to absorb enough water during the gelatinization process, and starch gelatinization levels are lower [37,38]. Therefore, PGA has a significant effect on the specific volume and staling of bread. Interestingly, the amount of water present during staling controls the retrogradation of wheat starch gels [20]. Hardness is primarily caused by the formation of cross-linked starch granules [39]. As steamed breads typically have a higher moisture level, this may accelerate starch recrystallization and contribute to higher hardness than that of baked breads [20].

## 5. Conclusions

This study compared the effects of three commercially available PGAs on mealworm-flour-formulated bread and steamed bread. Each PGA enhanced the quality of bread and steamed bread as demonstrated by significant increases in specific volume and moisture content, lower hardness, and slower staling. Overall, PGA from producer Q demonstrated the best effect on quality improvement of mealworm-formulated bread and steamed bread. The results of this study provide theoretical evidence for the development of insect-formulated flour-based products, and aid in the selection of appropriate PGAs for bread production.

## Figures and Tables

**Figure 1 foods-12-03641-f001:**
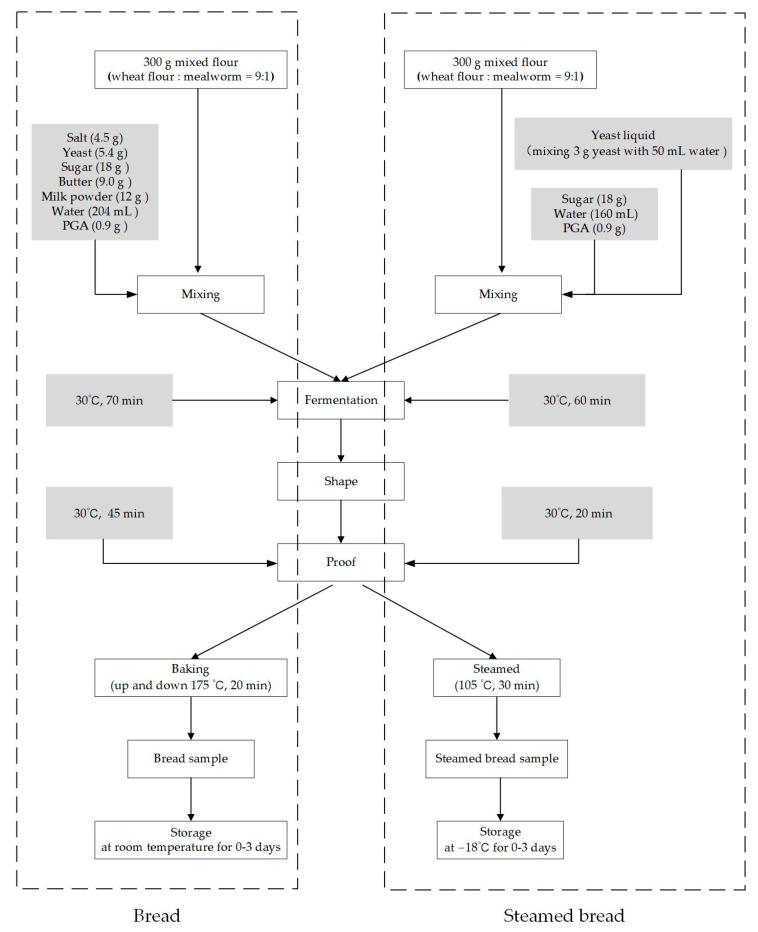
The main procedure for production of mealworm-flour-formulated bread and steamed bread with PGA added.

**Figure 2 foods-12-03641-f002:**
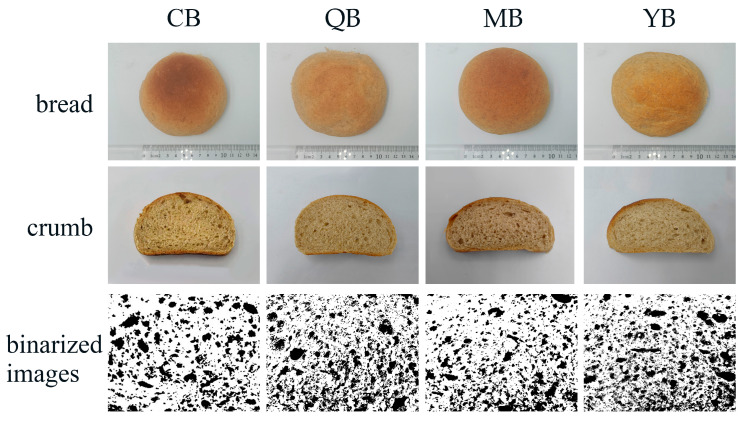
Visual appearance of the mealworm-flour-formulated bread and crumbs, and binarized images of the cross-section with different commercial PGAs. CB: Bread without PGA. QB: PGA from Qingdao Bright Moon Seaweed Group Co., Ltd. is added to the bread. MB: PGA from Shanghai Macklin Biochemical Co., Ltd. is added to the bread. YB: PGA from Yuanye Bio-Technology Co., Ltd. is added to the bread. PGA: propylene glycol alginate. PGA: propylene glycol alginate.

**Figure 3 foods-12-03641-f003:**
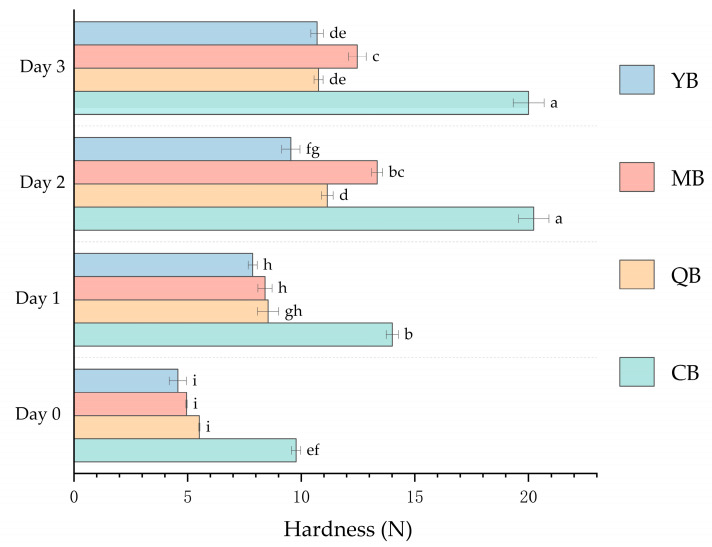
Effect of different commercial PGAs on the hardness of mealworm-flour-formulated bread during storage. CB: Bread without PGA. QB: PGA from Qingdao Bright Moon Seaweed Group Co., Ltd. Is added to the bread. MB: PGA from Shanghai Macklin Biochemical Co., Ltd. Is added to the bread. YB: PGA from Yuanye Bio-Technology Co., Ltd. Is added to the bread. PGA: propylene glycol alginate. Bars with different lower-case letters are significantly different at *p* < 0.05.

**Figure 4 foods-12-03641-f004:**
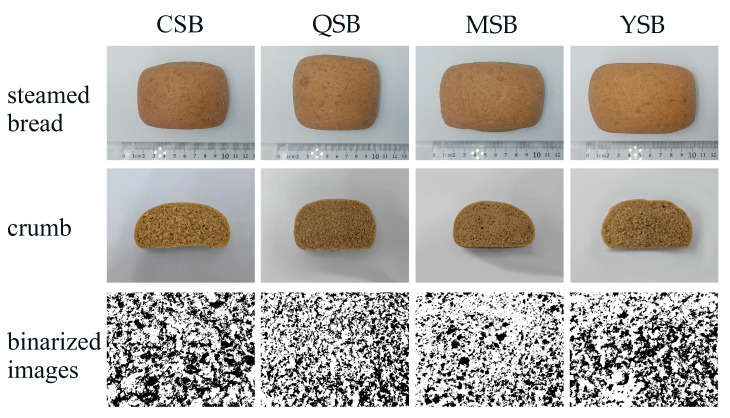
Visual appearance of the mealworm-flour-formulated steamed bread and crumbs, and binarized images of the cross-section with different commercial PGAs. CSB: Steamed bread without PGA. QSB: PGA from Qingdao Bright Moon Seaweed Group Co., Ltd. is added to the steamed bread. MSB: PGA from Shanghai Macklin Biochemical Co., Ltd. is added to the steamed bread. YSB: PGA from Yuanye Bio-Technology Co., Ltd. is added to the steamed bread. PGA: propylene glycol alginate.

**Figure 5 foods-12-03641-f005:**
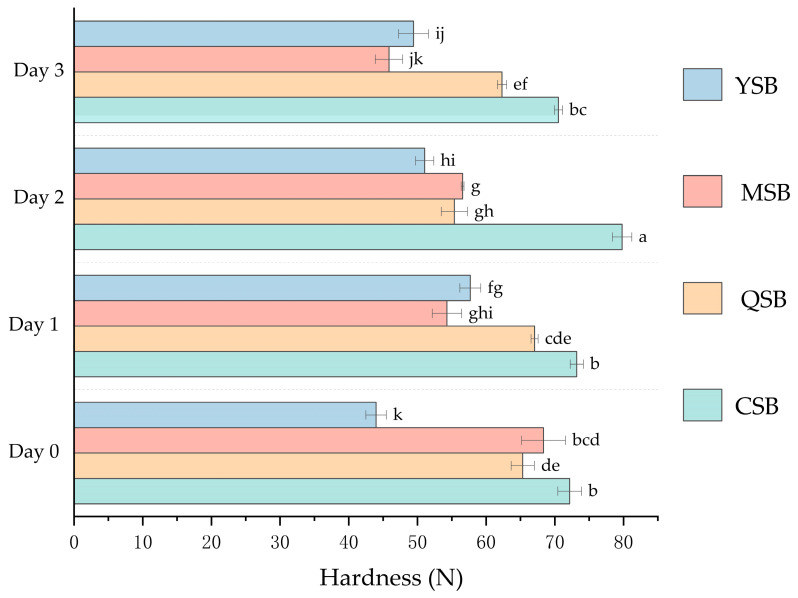
Effect of different commercial PGAs on the change of the hardness of mealworm-flour-formulated steamed bread during storage. CSB: Steamed bread without PGA. QSB: PGA from Qingdao Bright Moon Seaweed Group Co., Ltd. is added to the steamed bread. MSB: PGA from Shanghai Macklin Biochemical Co., Ltd. is added to the steamed bread. YSB: PGA from Yuanye Bio-Technology Co., Ltd. is added to the steamed bread. PGA: propylene glycol alginate. Bars with different lower-case letters are significantly different at *p* < 0.05.

**Table 1 foods-12-03641-t001:** Formulas for mealworm-flour-formulated bread with different commercial PGAs.

Sample Code	High-Gluten Wheat Flour (g)	Mealworm Flour (g)	Yeast (g)	Salt (g)	Sugar (g)	Milk Powder (g)	Butter (g)	Water ^a^ (mL)	Q (g)	M (g)	Y (g)
CB	270	30	5.4	4.5	18	12	9.0	204	-	-	-
QB	270	30	5.4	4.5	18	12	9.0	204	0.9	-	-
MB	270	30	5.4	4.5	18	12	9.0	204	-	0.9	-
YB	270	30	5.4	4.5	18	12	9.0	204	-	-	0.9

**^a^** The amount of water added is determined based on our previous experiments. CB: Bread without PGA. QB: PGA from Qingdao Bright Moon Seaweed Group Co., Ltd. is added to the bread. MB: PGA from Shanghai Macklin Biochemical Co., Ltd. is added to the bread. YB: PGA from Yuanye Bio-Technology Co., Ltd. is added to the bread. PGA: propylene glycol alginate.

**Table 2 foods-12-03641-t002:** Formulas for mealworm-formulated steamed bread with different commercial PGAs.

Sample Code	Whole Wheat Flour (g)	Mealworm Flour (g)	Yeast (g)	Sugar (g)	Water ^a^ (mL)	Q (g)	M (g)	Y (g)
CSB	270	30	3	18	210	-	-	-
QSB	270	30	3	18	210	0.9	-	-
MSB	270	30	3	18	210	-	0.9	-
YSB	270	30	3	18	210	-	-	0.9

**^a^** The amount of water added is determined based on our previous experiments. CSB: Steamed bread without PGA. QSB: PGA from Qingdao Bright Moon Seaweed Group Co., Ltd. is added to the steamed bread. MSB: PGA from Shanghai Macklin Biochemical Co., Ltd. is added to the steamed bread. YSB: PGA from Yuanye Bio-Technology Co., Ltd. is added to the steamed bread. PGA: propylene glycol alginate.

**Table 3 foods-12-03641-t003:** Color values (*L**, *a**, and *b**), specific volume, porosity, and moisture content of the mealworm-flour-formulated bread with different commercial PGAs.

	*L**	*a**	*b**	Specific Volume (mL/g)	Porosity (%)	Moisture Content (%)
CB	63.87 ± 3.12 ^a^	0.89 ± 0.18 ^b^	16.57 ± 0.74 ^a^	2.17 ± 0.25 ^b^	29.42 ± 1.50 ^b^	36.15 ± 0.48 ^b^
QB	63.82 ± 2.78 ^a^	1.56 ± 0.28 ^ab^	14.71 ± 1.87 ^a^	3.34 ± 0.03 ^a^	31.91 ± 0.30 ^a^	37.64 ± 1.14 ^a^
MB	62.13 ± 4.12 ^a^	1.84 ± 0.18 ^a^	15.09 ± 0.61 ^a^	3.40 ± 0.02 ^a^	28.23 ± 0.84 ^b^	36.69 ± 0.60 ^ab^
YB	67.79 ± 1.43 ^a^	1.47 ± 0.37 ^ab^	17.53 ± 2.29 ^a^	3.36± 0.04 ^a^	34.15 ± 0.54 ^a^	36.85 ± 0.29 ^ab^

Values in the same column followed by different superscripts are significantly different (*p* < 0.05). CB: Bread without PGA. QB: PGA from Qingdao Bright Moon Seaweed Group Co., Ltd. is added to the bread. MB: PGA from Shanghai Macklin Biochemical Co., Ltd. is added to the bread. YB: PGA from Yuanye Bio-Technology Co., Ltd. is added to the bread. PGA: propylene glycol alginate.

**Table 4 foods-12-03641-t004:** Effect of different commercial PGAs on the texture of the mealworm-flour-formulated bread during storage.

	Hardness (N)	Springiness	Chewiness (N)	Gumminess (N)	Cohesiveness	Resilience
Day 0						
CB	9.77 ± 0.21 ^ef^	0.88 ± 0.02 ^abc^	5.81 ± 0.17 ^cde^	6.64 ± 0.17 ^cdef^	0.68 ± 0.03 ^abcde^	0.33 ± 0.01 ^cdef^
QB	5.52 ± 0.03 ^i^	0.92 ± 0.03 ^a^	4.10 ± 0.49 ^ef^	4.44 ± 0.40 ^fgh^	0.80 ± 0.07 ^a^	0.47 ± 0.05 ^ab^
MB	4.95 ± 0.03 ^i^	0.92 ± 0.02 ^ab^	3.53 ± 0.16 ^ef^	3.85 ± 0.09 ^gh^	0.78 ± 0.02 ^ab^	0.48 ± 0.00 ^a^
YB	4.57 ± 0.38 ^i^	0.88 ± 0.00 ^abc^	2.90 ± 0.22 ^f^	3.28 ± 0.23 ^h^	0.72 ± 0.02 ^abcd^	0.38 ± 0.02 ^cde^
Day 1						
CB	14.01 ± 0.27 ^b^	0.84 ± 0.02 ^cd^	7.00 ± 0.78 ^bc^	8.34 ± 0.75 ^bc^	0.60 ± 0.06 ^cde^	0.26 ± 0.03 ^fg^
QB	8.54 ± 0.47 ^gh^	0.91 ± 0.01 ^ab^	5.77 ± 0.08 ^cde^	6.31 ± 0.05 ^cdef^	0.74 ± 0.04 ^abc^	0.39 ± 0.02 ^bcd^
MB	8.41 ± 0.31 ^h^	0.91 ± 0.02 ^ab^	5.42 ± 0.20 ^cde^	5.97 ± 0.32 ^defg^	0.71 ± 0.01 ^abcd^	0.40 ± 0.01 ^abc^
YB	7.86 ± 0.20 ^h^	0.87 ± 0.01 ^abcd^	4.48 ± 0.41 ^def^	5.18 ± 0.41 ^efgh^	0.66 ± 0.04 ^abcde^	0.31 ± 0.02 ^def^
Day 2						
CB	20.22 ± 0.67 ^a^	0.84 ± 0.05 ^cd^	10.62 ± 1.97 ^a^	12.65 ± 1.63 ^a^	0.63 ± 0.08 ^bcde^	0.27 ± 0.03 ^fg^
QB	11.15 ± 0.26 ^d^	0.87 ± 0.03 ^abcd^	5.60 ± 0.49 ^cde^	6.42 ± 0.37 ^cdef^	0.58 ± 0.04 ^de^	0.26 ± 0.03 ^fg^
MB	13.35 ± 0.24 ^bc^	0.90 ± 0.03 ^abc^	7.04 ± 0.24 ^bc^	7.86 ± 0.53 ^cd^	0.59 ± 0.03 ^cde^	0.30 ± 0.03 ^ef^
YB	9.54 ± 0.40 ^fg^	0.85 ± 0.03 ^bcd^	5.14 ± 0.63 ^cdef^	6.05 ± 0.67 ^cdefg^	0.63 ± 0.07 ^bcde^	0.27 ± 0.03 ^fg^
Day 3						
CB	20.01 ± 0.67 ^a^	0.81 ± 0.01 ^d^	8.44 ± 1.52 ^ab^	10.48 ± 1.85 ^ab^	0.52 ± 0.10 ^e^	0.21 ± 0.06 ^g^
QB	10.76 ± 0.20 ^de^	0.88 ± 0.01 ^abc^	6.42 ± 0.25 ^bcd^	7.29 ± 0.33 ^cde^	0.68 ± 0.03 ^abcde^	0.29 ± 0.01 ^efg^
MB	12.47 ± 0.39 ^c^	0.88 ± 0.01 ^abc^	7.00 ± 0.78 ^bc^	7.94 ± 0.92 ^cd^	0.64 ± 0.06 ^bcde^	0.31 ± 0.03 ^def^
YB	10.70 ± 0.27 ^de^	0.87 ± 0.02 ^abcd^	5.75 ± 0.74 ^cde^	6.65 ± 0.76 ^cdef^	0.62 ± 0.07 ^bcde^	0.25 ± 0.03 ^fg^

Values in the same column followed by different superscripts are significantly different (*p* < 0.05). CB: Bread without PGA. QB: PGA from Qingdao Bright Moon Seaweed Group Co., Ltd. Is added to the bread. MB: PGA from Shanghai Macklin Biochemical Co., Ltd. Is added to the bread. YB: PGA from Yuanye Bio-Technology Co., Ltd. Is added to the bread. PGA: propylene glycol alginate.

**Table 5 foods-12-03641-t005:** Color values (*L**, *a** and *b**), specific volume, porosity, and moisture content of the mealworm-flour-formulated steamed bread with different commercial PGAs.

	*L**	*a**	*b**	Specific Volume (mL/g)	Porosity (%)	Moisture Content (%)
CSB	56.75 ± 0.75 ^b^	4.96 ± 0.70 ^a^	22.51 ± 0.51 ^a^	1.72 ± 0.056 ^b^	42.16 ± 2.31 ^a^	45.47 ± 0.12 ^c^
QSB	61.55 ± 1.49 ^a^	5.64 ± 0.90 ^a^	26.13 ± 0.68 ^a^	2.23 ± 0.030 ^a^	38.98 ± 0.34 ^b^	46.90 ± 0.15 ^a^
MSB	60.10 ± 1.34 ^ab^	5.55 ± 0.46 ^a^	21.86 ± 1.59 ^b^	1.22 ± 0.038 ^c^	29.63 ± 0.46 ^c^	46.42 ± 0.27 ^ab^
YSB	57.91 ± 1.72 ^b^	5.69 ± 0.02 ^a^	25.91 ± 0.85 ^a^	1.51 ± 0.16 ^b^	39.7 ± 0.79 ^b^	46.14 ± 0.57 ^bc^

Values in the same column followed by different superscripts are significantly different (*p* < 0.05). CSB: Steamed bread without PGA. QSB: PGA from Qingdao Bright Moon Seaweed Group Co., Ltd. is added to the steamed bread. MSB: PGA from Shanghai Macklin Biochemical Co., Ltd. is added to the steamed bread. YSB: PGA from Yuanye Bio-Technology Co., Ltd. is added to the steamed bread. PGA: propylene glycol alginate.

**Table 6 foods-12-03641-t006:** Effect of different commercial PGAs on texture the mealworm-flour-formulated steamed bread during storage.

	Hardness (N)	Springiness	Chewiness (N)	Gumminess (N)	Cohesiveness	Resilience
Day 0						
CSB	72.17 ± 1.72 ^b^	0.70 ± 0.01 ^g^	28.73 ± 0.69 ^efgh^	40.87 ± 0.33 ^abcde^	0.57 ± 0.01 ^b^	0.22 ± 0.00 ^g^
QSB	65.33 ± 1.69 ^de^	0.88 ± 0.01 ^abcd^	43.56 ± 6.37 ^ab^	49.48 ± 6.65 ^ab^	0.76 ± 0.08 ^a^	0.39 ± 0.02 ^abcd^
MSB	68.35 ± 3.21 ^bcd^	0.91 ± 0.02 ^a^	45.27 ± 3.01 ^a^	49.71 ± 2.45 ^ab^	0.73 ± 0.06 ^ab^	0.44 ± 0.04 ^a^
YSB	43.97 ± 1.48 ^k^	0.76 ± 0.01 ^fg^	21.91 ± 3.23 ^h^	28.86 ± 3.80 ^f^	0.66 ± 0.06 ^ab^	0.27 ± 0.03 ^efg^
Day 1						
CSB	73.19 ± 0.97 ^b^	0.81 ± 0.03 ^def^	41.58 ± 3.75 ^abc^	51.30 ± 5.86 ^a^	0.70 ± 0.09 ^ab^	0.28 ± 0.04 ^efg^
QSB	67.05 ± 0.52 ^cde^	0.87 ± 0.03 ^abcd^	39.25 ± 1.62 ^abcd^	44.91 ± 0.54 ^abc^	0.67 ± 0.05 ^ab^	0.35 ± 0.01 ^bcde^
MSB	54.29 ± 2.14 ^ghi^	0.91 ± 0.01 ^a^	36.56 ± 0.89 ^abcdef^	40.17 ± 1.36 ^bcde^	0.74 ± 0.06 ^ab^	0.42 ± 0.04 ^ab^
YSB	57.69 ± 1.53 ^fg^	0.82 ± 0.03 ^cdef^	33.37 ± 5.39 ^cdefg^	40.57 ± 5.06 ^bcde^	0.70 ± 0.09 ^ab^	0.31 ± 0.04 ^def^
Day 2						
CSB	79.80 ± 1.41 ^a^	0.84 ± 0.01 ^bcde^	42.79 ± 0.81 ^abc^	51.27 ± 1.20 ^a^	0.64 ± 0.01 ^ab^	0.29 ± 0.01 ^efg^
QSB	55.37 ± 1.90 ^gh^	0.88 ± 0.02 ^abcd^	34.00 ± 2.03 ^bcdefg^	38.70 ± 2.02 ^cdef^	0.70 ± 0.06 ^ab^	0.33 ± 0.02 ^cde^
MSB	56.57 ± 0.21 ^g^	0.88 ± 0.04 ^abc^	37.82 ± 4.10 ^abcde^	42.79 ± 3.66 ^abcd^	0.76 ± 0.06 ^a^	0.45 ± 0.04 ^a^
YSB	51.05 ± 1.33 ^hi^	0.77 ± 0.04 ^ef^	24.75 ± 4.24 ^gh^	31.92 ± 3.94 ^ef^	0.63 ± 0.08 ^ab^	0.25 ± 0.03 ^fg^
Day 3						
CSB	70.53 ± 0.60 ^bc^	0.81 ± 0.03 ^def^	37.61 ± 2.10 ^abcde^	46.33 ± 3.56 ^abc^	0.66 ± 0.06 ^ab^	0.27 ± 0.01 ^efg^
QSB	62.31 ± 0.68 ^ef^	0.82 ± 0.03 ^cdef^	34.01 ± 4.27 ^bcdefg^	41.19 ± 3.87 ^abcde^	0.66 ± 0.07 ^ab^	0.32 ± 0.03 ^def^
MSB	45.85 ± 1.99 ^jk^	0.89 ± 0.01 ^ab^	29.28 ± 1.50 ^defgh^	32.80 ± 1.72 ^def^	0.72 ± 0.01 ^ab^	0.41 ± 0.02 ^abc^
YSB	49.43 ± 2.18 ^ij^	0.84 ± 0.01 ^bcd^	27.60 ± 1.53 ^fgh^	32.82 ± 1.69 ^def^	0.66 ± 0.01 ^ab^	0.30 ± 0.01 ^ef^

Values in the same column followed by different superscripts are significantly different (*p* < 0.05). CSB: Steamed bread without PGA. QSB: PGA from Qingdao Bright Moon Seaweed Group Co., Ltd. is added to the steamed bread. MSB: PGA from Shanghai Macklin Biochemical Co., Ltd. is added to the steamed bread. YSB: PGA from Yuanye Bio-Technology Co., Ltd. is added to the steamed bread. PGA: propylene glycol alginate.

## Data Availability

Data are not available in public datasets; please contact the authors.

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
