# Peer review of "Effect of Adding Different Commercial Propylene Glycol Alginates on the Properties of Mealworm-Flour-Formulated Bread and Steamed Bread"

_foods, 2023, doi:10.3390/foods12193641_

Round 1

Reviewer 1 Report (Previous Reviewer 1)

The study is improved according to my suggestions and now is ready for publication

Reviewer 2 Report (Previous Reviewer 2)

The article has been improved sufficiently. 

Reviewer 3 Report (Previous Reviewer 3)

As I wrote earlier, due to the use of insects as a source of protein and other ingredients, the manuscript is valuable scientifically, and even more so practically. A novelty is the recipe with the addition of mealworm powder with the addition of propylene glycol alginate to enhance the texture and other properties of baked and steamed bread.

After taking into account all comments from all reviewers, the article is improved and suitable for printing. I hope that the still existing editing errors, e.g. missing spaces, will probably be corrected, at least in proofreading.

The quality of the English language is good.

This manuscript is a resubmission of an earlier submission. The following is a list of the peer review reports and author responses from that submission.

Round 1

Reviewer 1 Report

The study is about the effect of PGA in bread and steamed bread with mealworm. The study is well organized and the results are clearly stated.

Information and data of the three PGA used for the bread formulations should be given, regarding its characteristics. Were they different in any way?

Moisture loss should be measured during storage to better explain hardening. 

Reviewer 2 Report

The article concerns an important experiment testing the effect of PGA on the quality characteristics of wheat typical bread and steamed bread made with 10% mealworm powder mixtures.

Since the journal is international, and there is currently a clean-label trend in the world, it is important to discuss what regulation looks like allowing PGA to enter markets in different economic zones.

However, before one can consider the correctness of the presentation of the results, in this article, it is necessary to point out the basic assumption of the research experiment - reproducibility.

The lack of information on the detailed composition of individual PGA products, as well as the lack of information on the composition of mealworm powder, makes it impossible, firstly, to reproduce the experiment, and secondly, to verify the correctness of the inference in the discussion.

Also, the methods should be described in more detail - this applies to pt. 2.5. and 2.8.1. where the sample diameter is missing. 

There is also no indication of the dimensions of the bread that was obtained.

Only after correcting these deficiencies, you can resubmit the article. 

Reviewer 3 Report

The manuscript is valuable scientifically, even more practically, because, in recent years, the use of insects as a source of protein and other ingredients has been increasingly considered around the world. In addition, the recipe with the addition of mealworm powder has been refined with the addition of propylene glycol alginate to enhance the texture and other properties of baked and steamed bread.

The manuscript is well written, structured, and supported by a discussion taking into account well-chosen literature sources, including those from recent years. The methodology is well described, and the results are presented in charts and tables, including statistical analysis. However, the manuscript needs re-reading to correct the style of some sentences and to remove minor editing errors, especially missing spaces.

I would suggest improving the abstract to make it clear how much mealworm powder was added to the bread recipe, and how much and in what form propylene glycol alginate was added. The influence of commercial preparations of propylene glycol alginate was studied in the manuscript, but their chemical composition was not given. What was the difference between these preparations?

I suggest checking and possibly correcting the information about the replacement of traditional flour with mealworm powder with propylene glycol alginate or the addition of this powder throughout the manuscript. Already the title is a bit misleading, especially: „properties of mealworm powder formulated bread”.

Similarly, e.g. Caption of Figure 1 “The mealworm formulated bread and steamed bread main making procedure”

Line 164: “..the Tukey test was carried out to separate means at a significance level of 164 95% (p < 0.05)” – I propose to correct this sentence, it is usually about separating homogeneous groups and not separating averages.

The manuscript needs re-reading to correct the style of some sentences and to remove minor editing errors, especially missing spaces.